# Smart Automation for Production of Panchagavya Natural Fertilizer

**Sumathi V.** [1,*] and **Mohamed Abdullah J.** [2]

1 Centre for Automation, School of Electrical Engineering, VIT, Chennai 600127, Tamil Nadu, India
2 School of Electrical Engineering, VIT, Chennai 600127, Tamil Nadu, India
* Correspondence: vsumathi@vit.ac.in

**Abstract:** Modern agricultural farming techniques employ the usage of chemical supplements to improve crop yield in terms of quantity and quality. This practice has brought down the fertility of the soil and has led to secondary impacts and necessitates a significant financial investment. Awareness of the side effects of artificially enriched food has made people move towards organically grown food, and the consumption has also increased significantly. One of the ancient organic fertilizers used in India is panchagavya. As the name implies, pancha means five and gavya means cow. The five products of the cow are combined as per the compositions and procedure described in the literature, to provide economical and meaningful value to organic farming. The objective of this work is to design, develop, and implement an automated system to manufacture panchagavya with reduced operator assistance. The system implements an ATmega 328 microcontroller to automate the entire process by interfacing sensors such as pH, moisture, temperature, and pressure. The system is also provided with a SIM900A GSM modem to provide information to the user regarding the status of the process. The developed pilot scale design discussed in this work has several advantages in the world of farming technologies in terms of enriching the soil, thereby improving the crop yield. This technology will benefit the farmers as this natural fertilizer can be mass-produced and turn them into entrepreneurs, which benefits society at large.

**Keywords:** panchagavya; organic fertilizer; liquid fertilizer; automated fertilizer production; drip irrigation system; automated irrigation



## 1. Introduction

The use of chemicals in farming improves the crop yield, but this has led to secondary effects as well, such as drastic reduction in the fertility of the soil and health-related issues in humans. These factors have presently led to adapting the age-old organic farming system. An ideal organic farming system enhances the process of nutrient cycling and reduces the usage of external inputs [1]. The most widely practiced organic farming system in ancient India uses panchagavya, vermicompost, and farmyard manure for nutrient management and soil enrichment. Panchagavya, derived from ancient Indian times, has always been treasured as a valuable possession of the country. It has been used for generations in enriching and improving the soil, and provides health benefits for both humans and animals. It has proved to be successful within a short duration. Among several benefits of this natural fertilizer, the agricultural era has benefited the most. The five cow products use three dairy products: cow milk, curd, and ghee, which is mixed with cow dung and urine [2]. The usage of chemicals adopted during the industrial revolution has led to the diminishing use of the natural fertilizer panchagavya. A study carried out in Andhra Pradesh, India [3], reveals the benefits of panchagavya as a fertilizer and also a seed storage treatment entity. It also characterizes three different organic preparations concerning microbiological aspects and the impacts made on crop growth and yield given in Figure 1.

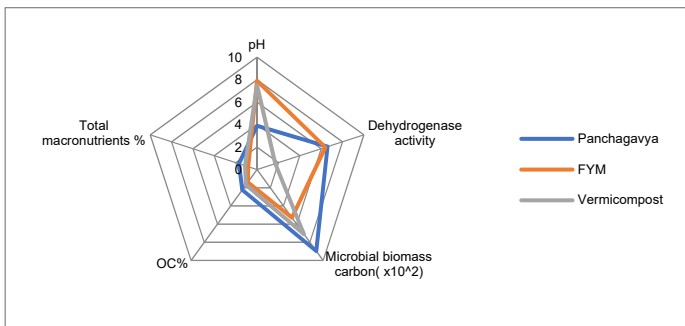

**Figure 1.** Microbiological quality and impacts of different organic fertilizers [3].

The use of panchagavya in a field experiment conducted on brinjal (*Solanum melongena*) in 2008–2009 by the University of Agricultural Sciences, Dharwad (Karnataka), India, has exhibited good results in brinjal yield. The nutrient uptake has been reported to be N (92.86 kg ha$^{-1}$), P (22.16 kg ha$^{-1}$), K (110.62 kg ha$^{-1}$), and S (29.48 kg ha$^{-1}$) [4]. Apart from several advantages related to plantation and crop yield, the most specific application is in the use of seeds or seedling treatment and can be used as a catalyst for organic manure for speeding up its decomposition [5]. Experienced personnel use panchagavya as a medical supplement to treat people for various diseases and deficiencies. Though the different applications listed require a 3% concentration of panchagavya, some plants, such as cashew rootstocks, require a 5% concentration for beneficial growth parameters and graft success [6]. Effects of the application of panchagavya in the form of seed treatment and foliar spray to southern sunn-hemp mosaic virus-infected sun hemp plants have been reported [7]. Since panchagavya not only holds itself useful in the field of agriculture but also plays a vital role in the health of humans and animals, researchers can bring out the possible benefits of panchagavya as a medicine [8–10].

The conventional method in practice for the preparation of panchagavya involves the mixing of cow dung and ghee thoroughly in a mud pot (or any other vessel except metal containers as panchagavya is acidic). This mixture is stirred every 12 h for 3 days. Once the process is completed, the remaining three ingredients are added, and again, the mixture is stirred thoroughly. At this stage, any one of the catalysts may be added, such as sugarcane juice and jaggery for speeding up the process. This mixture is stirred every 12 h for 15 days, and it is ready for use after the 18th day. The entire process is carried out manually, which requires physical effort and tolerance of uncomfortable smell and timekeeping. The ingredient proportions mentioned in Table 1 yield approximately 20 L of panchagavya, as reported in the literature.

**Table 1.** Composition of panchagavya.

| S. No. | Ingredients Required for the Preparation of Panchagavya | Composition for Making 20 L of Panchagavya |
|---|---|---|
| 1 | Cow dung | 5 kg |
| 2 | Cow urine | 3 L |
| 3 | Cow milk | 2 L |
| 4 | Cow ghee (clarified butter) | 500 g |
| 5 | Curd | 2 L |

This paper proposes the implementation of a new method for modern organic farming techniques, bringing sustainable changes in the field of agriculture technology by developing a cost-effective design for the production of panchagavya with minimal human effort. An attempt is made to design, develop, and automate a microcontroller-based system in pilot scale to overcome the disadvantages of the manual method in practice. The proposed work, when implemented in the irrigation field, uses a fertigation technique, that is, mixing of liquid fertilizer with irrigation water [11] and with more focus on applying in the surface

irrigation method [12]. The first section of the article discusses the methodology involved in the design and development of the natural fertilizer panchagavya's preparation. The scaled-down values for pilot-scale implementation is also discussed. The next section highlights the working of the developed system, followed by the design implementation of the experimental setup. Finally, the results and conclusion of this research are discussed.

## 2. Methodology

The pilot design shown in Figure 2 for the preparation of panchagavya consists of three stages, which yields 1 L of panchagavya.

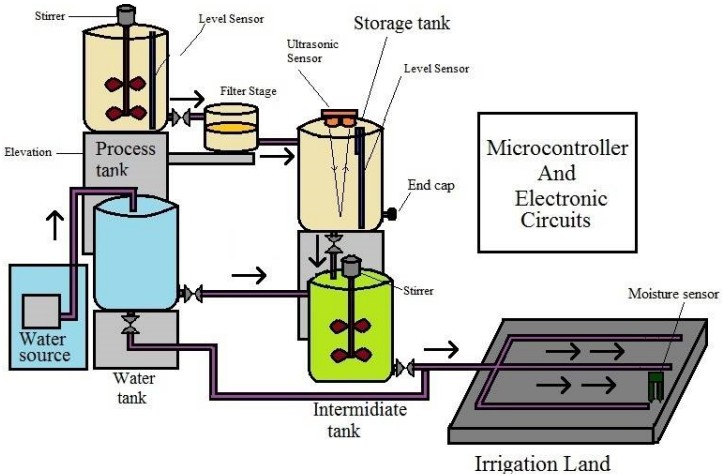

**Figure 2.** Block diagram of the designed system.

The block diagram shown above consists of the process tank, storage tank, water tank, and intermediate tank. The process tank is attached to a feeder, blender assembly, level sensor, and GSM module. The blender assembly is mounted vertically on top of the process tank. The assembly consists of a DC motor coupled with an elongated shaft and blades for rotary action. The solenoid valves used in this system have a larger outlet orifice compared with other valves used since unfiltered panchagavya passes through this valve. The end-to-user communication is carried out using the GSM module. The storage tank is used for storing the liquid fertilizer. Since the panchagavya is acidic in nature and cannot be directly irrigated into the field, the liquid fertilizer is mixed with water from the water tank and stored in the intermediate tank. The Atmel ATmega 328 microcontroller is integrated into the system to monitor the physical parameters of the soil, such as moisture and pH.

## 3. Working of the Desired System

The ingredients used for the preparation of 1 L of panchagavya have been scaled from Table 1 and are prescribed in Table 2 and placed in the process tank. The process tank's inner layer is made of a nonmetallic body, and the input is loaded through a hopper feeder placed above the top lid of the tank and is sealed. The pH and the temperature values are displayed as sensed by the pH sensor and temperature sensor inside the process tank. A horizontally placed motor on top of the process tank acts as the blender to mix the composition evenly at regular intervals. Continuous mixing of the composition at regular intervals aids the culturing process and avoids sedimentation. Once the process ends, the prepared mixture is processed, the microprocessor sends a command to the outlet solenoid valve placed in the process tank to open, and the liquid flows gradually to the filter tank. In the filter tank, using multiple layers of filter, the mixture is filtered and the liquid fertilizer is separated from the residue and passed into storage tank. The storage tank is monitored by using a pH sensor and level sensor given in Table 3. The level of the liquid fertilizer in the storage tank is measured. At every stage of the process, the level in each of the tanks

is monitored by the level sensor placed in the tanks interfaced with the microcontroller. The available panchagavya in the storage tank has to be mixed appropriately with water. The water storage system for irrigation is controlled by the microcontroller by turning ON and OFF the pump using the relay. The suggested ratio of mixing panchagavya with water is 30:70, which is stored in the intermediate tank for irrigation. The sensors interfaced to every stage of the process is controlled by the Atmel ATmega 328 microcontroller.

**Table 2.** Scaled-down ingredients.

| S. No. | Ingredients | Quantity |
|--------|-------------|----------|
| 1 | Fresh cow dung | 500 g |
| 2 | Cow urine | 300 mL |
| 3 | Cow milk | 200 mL |
| 4 | Cow curd | 200 mL |
| 5 | Cow ghee | 50 g |
| 6 | Ripe banana | 1 no |

**Table 3.** Ultrasonic-level sensor readings.

| Level of Panchagavyam | Level Sensor Readings (in cm) |
|-----------------------|-------------------------------|
| Empty | Above 16 |
| Low | 13–16 |
| Medium | 10–12 |
| High | 7–9 |

## 4. Experimental Setup of the Implemented Design

The liquid fertilizer ready to be fed into the field cannot be transferred as such. A pilot study was conducted to farmers from a small village in Kandigai, Chennai, Tamil Nadu, India, on the time constraints followed in irrigating the field when drip irrigation is adopted. Based on the inputs received from them, it is ideal to supply water for 4 h using the drip irrigation method to irrigate the field. The work focuses on transferring the liquid fertilizer to the soil by drip irrigation, and thus, the 4 hours of supplying water to the field by drip irrigation is divided into four slots. As per these proportionated time slots, an algorithm for drip irrigation is designed. In the first slot, only water is fed to the crops. After an hour, the value of the pH sensor readings is considered to supply the liquid fertilizer; if the pH value is within the nominal range, the microprocessor turns ON the intermediate tank valve to open. The reason behind not feeding the panchagavya-mixed water directly to the first time slot is to avoid the supplied enriched water from directly being absorbed by the soil, and the crops will be left with very few nutrients. Two slots are used to irrigate the land with the liquid fertilizer. Finally, when the countdown timer is left with the remaining 1 hour, the intermediate tank valve is closed and water is fed to the irrigation system. The algorithm for drip irrigation is shown in Figure 3.

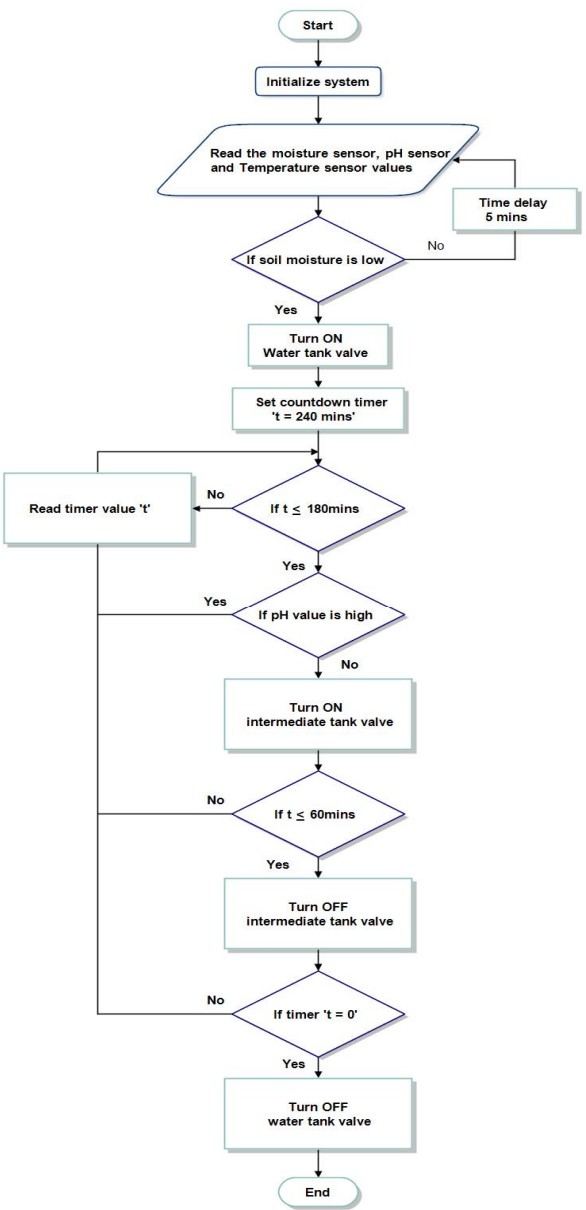

**Figure 3.** Flowchart of the automated drip irrigation process.

## 5. Results and Discussion

The proposed work in pilot scale is implemented, and the setup is shown in Figure 4. This model is designed to produce 1 L of panchagavya. For the requirement of a nonmetallic tank surface, all the tanks used in this model are chosen to be plastic-based containers. The introduction of preparation and irrigation modules has led to the inclusion of two microcontrollers. Two Atmel ATmega 328 microcontrollers are used, one as master and the other as slave. Each microcontroller carries 14 digital input/output pins and 6 analog inputs, which support the signals from level sensors, pH sensors, moisture sensors, and ultrasonic sensors and support communication through the GSM module given in Figure 4a.

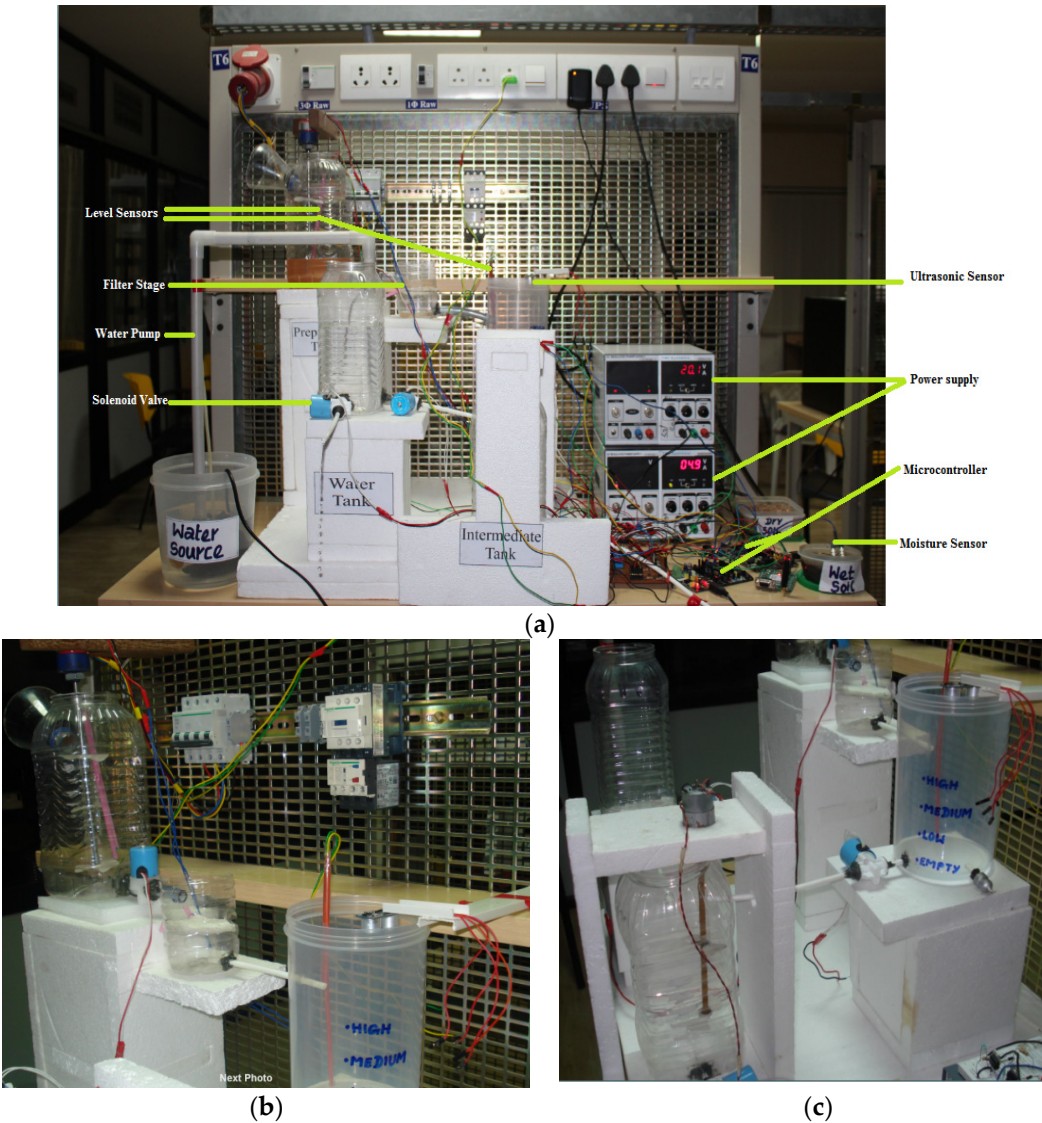

**Figure 4.** (**a**) Hardware setup of the proposed pilot scale model, (**b**) implementation of the preparation module, and (**c**) implementation of the irrigation module.

Since the production is planned for a small scale, a DC motor with high torque and low rpm is used for blender operation. A driver circuit is included to operate the DC motor interfaced to the microcontroller. Solenoid valves used in various levels interconnect different stages of the tank, and the HR1-type solenoid valve of a different size is used in this model. A conductance level sensor is used in both the preparation tank and the storage tank, which prevents overflowing given in Figure 4b,c. The working of this sensor is similar to the float-type sensor to indicate the brim level. An HC-SR04 model ultrasonic sensor is used in the storage tank to measure the level of the liquid fertilizer. A SIM900A module GSM modem is connected to the preparation to serve the purpose of communicating between the processing end and user end. The start and end of the blender operation and irrigation process at regular intervals are notified to the user by a text message and sent through the SMS service.

The model has been successfully implemented for the automated preparation of panchagavya with a surface drip irrigation system given in Figure 5a, and the installation of a moisture sensor is given in Figure 5b. Moisture sensor reading in voltage is shown in Figure 6. From this study, we understand that the positioning of a moisture sensor and pH sensor on the surface of the soil is to be wisely chosen, depending upon the crop's

requirement, and it is proven from [13] that it is ideal to measure under 10–15 cm from the soil surface. Depending upon the different crop needs, the type of irrigation with a liquid fertilizer is recommended as surface, subsurface, and root infiltration, as suggested in [12,14,15]. The drip irrigation systems are usually powered by a gravity tank for the flow of water to reach each node of the drip system. If the field to be covered is large, then the ordinary tank setup would not support the extent; hence, an intermediate pump would be required to cover the field. The sensors should be placed widely in the field to ensure that the region is properly covered for optimal utilization.

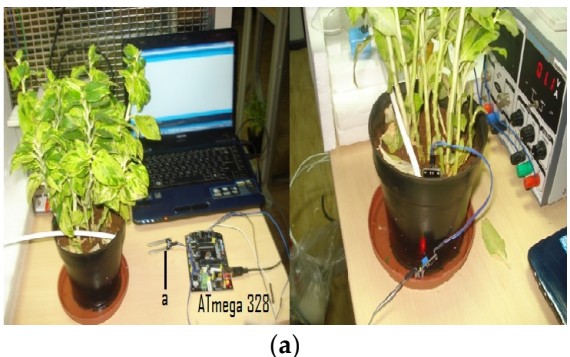

(**a**)

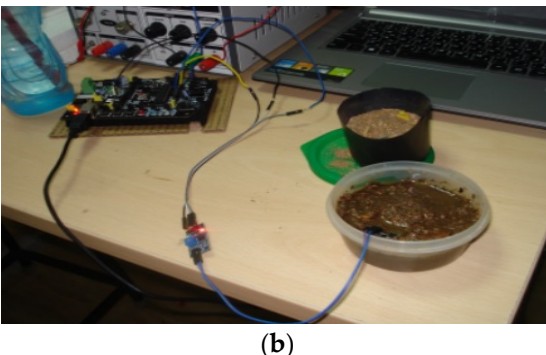

(**b**)

**Figure 5.** (**a**) Soil moisture sensor and pH sensor installed and tested; (**b**) soil moisture sensor placed in panchagavya.

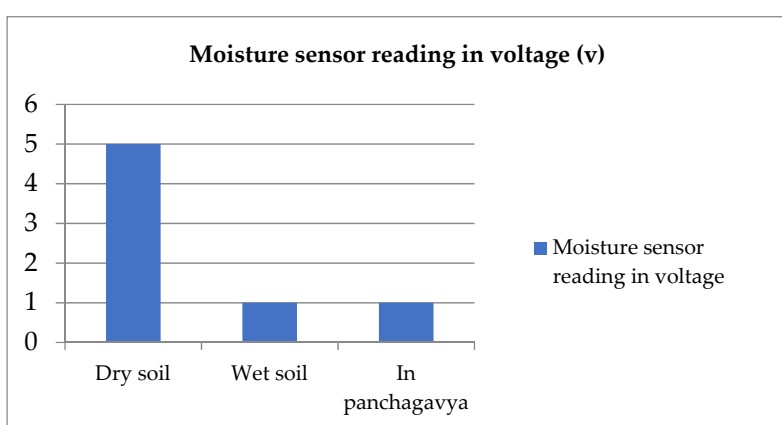

**Figure 6.** Moisture sensor reading in voltage.

## 6. Conclusions

The prototype of the ATmega 328 microcontroller-based automatic preparation of panchagavya natural fertilizer supported with drip irrigation is developed in pilot scale, and the results are reported. It is clear from the results that the panchagavya prepared by the automatic method has more advantage than the conventional method. In the preparation stage, the manual method requires much human effort, very tedious preparation, foul smell, and no information on the physical parameters of the liquid fertilizer; that is, moisture, pH, and so on are all overcome in the present development. The interfacing of pH sensors, moisture sensor, ultrasonic sensor, and GSM module gives immediate update on the process as the user is notified at every stage and, thus, eliminates the need for humans to be tied to the process. The developed prototype is low cost and, if implemented in a large scale, can be upgraded easily with minor modifications. The research concludes that if this method is adopted, the use of the diminishing natural fertilizer panchagavya can be brought to the limelight, which is much beneficial to the agricultural sector, improving more yield economically. Despite the several health benefits being reported in the literature, the farmers at large are benefitted, turning them into entrepreneurs and improving their livelihood.

**Funding:** The authors wish to thank VIT management for providing sufficient funds towards publication of the work.

**Conflicts of Interest:** The authors declare no conflict of interest.

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
