# Peer review of "Smart Automation for Production of Panchagavya Natural Fertilizer"

_agronomy, doi:10.3390/agronomy12123044_

Round 1

Reviewer 1 Report

The authors propose an automated irrigation system that manufactures Panchagavya with reduced operator assistance and appropriate irrigation of the agricultural land.
1) The introduction can be aligned with the authors' objective in this paper. For example, it is not clear the main contribution of the paper. What is the
contribution of this work to the precision irrigation area? Many solutions use an automated irrigation system.

2) The authors must improve the related papers. Few works were mentioned, and it is missing recent and related papers. This aspect also helps to understand the paper's contribution.

3) How did the authors validate the proposed system? And how about the collected data? The experiments and methodology and design experimental are not presented.  They must present results to support their conclusions. They must include results about the collected data and the actuator's processes.

Author Response

Author's response to Reviewer #1

  • Contributions have been revised.
  • References are updated
  • The experiments, methodology and design are presented in the attached manuscript. The results obtained from the study are also presented.

We are very thankful to the reviewer for the valuable suggestions to improve the quality of the paper.

Reviewer 2 Report

The manuscript can consider to correction as listed below

1.       The title is recast to be “ Smart Automation for Mass Production of Organic Fertilizer using five Products of Cow (Panchagavya)”

2.       In the Abstract line 24, an entrepreneurs not an entrepreneur as singular form (add s to the word)

3.       Where is the study area for this research? The study area where the research was carried out need to be mention in the manuscript

4.       After initial add dot. Sumathi V. and Muhammed Abdullah J.

5.       In line 74, the paragraph need to be back up with reference

6.       The picture supposed to be titled as Plate not figure

7.       Reference need to be cited for table 3.1

8.       The title captured mass production of Panchagavya while in the manuscript it found to be in a small scale

9.       It is suggested you should use graphs, or charts in presenting your findings

10.    In the list of references, reference 2 line 309, mention all the authors name in the article you cited. (Do not use et al., in the list of references)

11.    Make all the botanical name to be Italic ( e.g in line 310 Cajanus cajan L.)

Author Response

Author's response to Reviewer #2

  1. References are updated.
  2. The correction is carried out accordingly.
  3. The study of this research is presently mentioned in the corrected manuscript.
  4. The corrections are carried out as advised.
  5. The necessary corrections have been incorporated.
  6. The change is noted and corrected accordingly.
  7. Correction is done for this query.
  8. Title is modified as advised and can be seen in the attached corrected manuscript.
  9. Graphs are included in the corrected manuscript.
  10. Corrections are done accordingly.
  11. Corrections are done accordingly.

We are very thankful to the reviewer for the valuable suggestions to improve the quality of the paper.

Reviewer 3 Report

General:

Congratulations for Interdisciplinary, interesting, and actual topic, and results! The proposals are interesting for practical use, especially for automated system that manufactures Panchagavya!

The manuscript brings some interesting contributions in smart agriculture topic, for integrate organic farming system with a smart solution for monitoring the automated irrigation system.

Generally, I agree on publication.

However, there are some issues that must be addressed to improve the overall communication of author`s work:

·       Please, use the SI units (example: one part mg, another kg, etc. Use only kg and multiply factor).

·       Some parts need to be reformulated.

Title, abstract and paper content can be more correlate!

Abstract

Objective can be more expressive relative to title; automation usually reduce operator assistance (Rows 16-18).

Introduction

Analyze is clear, but, sometimes too comprehensive to the field. Must be presented relevant aspects in correlation with title and objectives.

In this area are presented the paper’s aim and objectives.

Please, use the SI units! (1litre =m3x10-3).

Experimental setup

The readers should find here (only and exclusively) detailed description of all your procedures (step by step), describe each equipment and part (if possible, all types, not only here and there, and processing parameters), and method (author, process conditions) used, anybody who reads this chapter should be able to repeat your methods and obtain the same results.

Automated process

Automated process means to implement a semi-automated, low-cost system!!! (Row 181-182).

Give more details about parts and equipment and write more technically (always provide corresponding numbers) and be more straightforward (condensate your text, remove ballast phrases and cliche).

Some numerical results during tests are missing. (If not, must precise the TRL level)

Implementation and discussion

The same observations as before.

Conclusion

Conclusions must be consistent with the evidence and arguments presented. Do not present revelations that are limited to your case study, provide deeper synthesis of your findings and present new revelations (use references in the same frame of results).

Author Response

Author's response to Reviewer #3

Abstract has been corrected and modified in the corrected manuscript.

Introduction: The objective is defined in the corrected manuscript.

Automation Process: All the necessary corrections have been done in the corrected manuscript.

Implementation and Discussion: It is discussed clearly in the corrected manuscript.

Conclusion: It is enriched in the corrected manuscript.

We are very thankful to the reviewer for the valuable suggestions to improve the quality of the paper.

Reviewer 4 Report

The authors present a proposal to design and implement an automated system for producing Panchagavya as a natural fertilizer. The comments and improvements required for the work presented are presented below.

The Introduction should be summarized, give the context, introduce the problem, and state the objective and the hypothesis (or research question). I recommend adding a paragraph at the end explaining the article's structure.

The elements that are left out of the Introduction should be part of a new section of Background or general concepts, which shows a synthesis of the state of the art and main concepts considered in the proposal (show more references regarding the topic).

If there are any, consider a new section giving a synthesis of works related to the authors' proposal, showing the similarities and differences and the contribution of the same.

The current section 2, considering the addition of the previously mentioned sections, should define, before the experimental configuration, the design or methodological framework that supports the proposal. From a technical perspective, it should be made clear why this configuration and the components selected; otherwise, the work, beyond its scientific contribution, could be seen as an experience report. The authors should provide the details that would eventually allow for a replication study.

Shouldn't the design of the automated process, which gives title to section 3, be part of the design of the proposal, and its result be presented in a results section?

In sections 2 to 4, the authors explain the design and implementation of the system. This explanation includes its components and interaction, but there are no detailed results of its use or the quality of the final product obtained. From the above, it could be established that the result presented is enough to be considered conceptual proof since it shows that the proposal is feasible but nothing more.

The discussion should be presented as an independent section, where the authors critically analyze the results obtained, verifying the fulfillment of the objectives and answer to the research question posed. They are also expected to compare with other existing proposals and discuss them. On the other hand, in this same section, they should consider the main weaknesses and threats to the results' validity.

OBS:

- Correct some typos.

- The authors should consider a broader and more updated set of bibliographic references. Sixty percent of the current references are more than five years old since their publication date.

Author Response

Author's response to Reviewer #4

The corrected manuscript addresses the review comments on summarizing the Introduction, Introduction of the problem, stating the objectives. Detailed results have been included with the experimental set up and explanation of its  implementation and results presented in the result section also.

Corrections of typos have been carried out.

Recent references have also been updated.

We are very thankful to the reviewer for the valuable suggestions to improve the quality of the paper.

Round 2

Reviewer 1 Report

The authors improve the contribution explanation, methodology, and results. Although they updated some references, they must enhance the introduction and improve the related work. I suggest a comparative table highlighting their main differentials related to the other works.

Author Response

This approach is the first of its kind which develops an automated system for production of panchagavya natural fertiliser which has several benefits from agricultural to health sector. The developed system overcomes the disadvantage of the current manual process in practise. The authors wish to convey that there are no similar approach and this design stands to be novel and unique.

We are very thankful to the reviewer for the valuable suggestions made to improve the quality of the paper.

Reviewer 2 Report

The manuscript has been sufficiently improved to warrant publication.

Author Response

The Authors wish to thank the reviewer for the positive reply and valuable suggestions made to improve the quality of the paper.

Reviewer 4 Report

The authors present an improved proposal to design and implement an automated system for producing Panchagavya as a natural fertilizer. However, not all the comments and improvements required for the work have been considered. 

The current version of the paper gets very difficult to see the required changes. On the other hand, the authors' answers are very concise about the improvements.

The Introduction has been summarized, but it should be more specific about giving the context, introducing the problem, and stating the objective and the hypothesis (or research question). I do not see a final paragraph explaining the article's structure.

The new section asked about Background, or general concepts, was not considered. Then there is no synthesis of the state of the art and main concepts considered in the proposal.

There is no explanation for not considering a section synthesizing related work to the authors' proposal, showing the similarities and differences and the contribution of the same.

The current section 2 describes the methodology but does not explain the design or methodological framework that supports the proposal. The configuration and the components selected are not justified; otherwise, the work, beyond its scientific contribution, could be seen as an experience report. The authors should provide the details that would eventually allow for a replication study.

The results & discussion are independent, but the authors do not critically analyze the results obtained, verifying the fulfillment of the objectives and answer to the research question. There is no comparison with other existing proposals. Also, the main weaknesses and threats to the results' validity were not presented.

OBS:

- The typos were corrected.

- The bibliographic references were updated.

Author Response

The current version of the paper gets very difficult to see the required changes. On the other hand, the authors' answers are very concise about the improvements.

Author’s response: 

The work is presented concisely due to the word count as required by the journal for communication article.

The Introduction has been summarized, but it should be more specific about giving the context, introducing the problem, and stating the objective and the hypothesis (or research question). I do not see a final paragraph explaining the article's structure.

Author’s response: 

In the introduction the problem statement is defined, the objective and hypothesis is presented and the article’s structure is included.

The new section asked about Background, or general concepts, was not considered. Then there is no synthesis of the state of the art and main concepts considered in the proposal.

There is no explanation for not considering a section synthesizing related work to the authors proposal, showing the similarities and differences and the contribution of the same.

Author’s response: 

This approach is the first of its kind which develops an automated system for production of panchagavya natural fertiliser which has several benefits from agricultural to health sector. The developed system overcomes the disadvantage of the current manual process in practise. The authors wish to convey that there are no similar approach and this design stands to be novel and unique.

The current section 2 describes the methodology but does not explain the design or methodological framework that supports the proposal. The configuration and the components selected are not justified; otherwise, the work, beyond its scientific contribution, could be seen as an experience report. The authors should provide the details that would eventually allow for a replication study.

Author’s response: 

The methodology and the working clearly explain the selection of the various tanks, the solenoid valves, sensors, controllers and the communication protocol for the design of automated preparation of panchagavya and the transferring of liquid fertilizer by drip irrigation technique. A pilot study on the manual process was conducted before implementation of the present design. Based on the study conducted the authors have designed the various tanks, selection of solenoid valves, sensors, controllers and the communication protocol which are easily accessible for the farmers and have been discussed in the section methodology, working of the desired system and experimental setup.

The results & discussion are independent, but the authors do not critically analyze the results obtained, verifying the fulfilment of the objectives and answer to the research question. There is no comparison with other existing proposals. Also, the main weaknesses and threats to the results' validity were not presented.

Author’s response: 

The weakness and threats of the manual process is overcome in the automated process and it is validated by using the sensor readings.
